# Reproducibility Study of "Learning Perturbations to Explain Time Series Predictions"

**Jiapeng Fan**[*]  **Luke Cadigan**[*]  **Paulius Skaisgiris**[*]  **Sebastian Arias**[*]

**Reviewed on OpenReview:** `https://openreview.net/forum?id=fCNqD2IuoD`

## Abstract

In this work, we attempt to reproduce the results of Enguehard (2023), which introduced ExtremalMask, a mask-based perturbation method for explaining time series data. We investigated the key claims of this paper, namely that (1) the model outperformed other models in several key metrics on both synthetic and real data, and (2) the model performed better when using the loss function of the preservation game relative to that of the deletion game. Although discrepancies exist, our results generally support the core of the original paper's conclusions. Next, we interpret ExtremalMask's outputs using new visualizations and metrics and discuss the insights each interpretation provides. Finally, we test whether ExtremalMask create out of distribution samples, and found the model does not exhibit this flaw on our tested synthetic dataset. Overall, our results support and add nuance to the original paper's findings. Code available at [this link](#).

## 1 Introduction

Machine learning (ML) methods are commonly applied to analyze time series data in critical situations, such as predicting patient survival using vital sign readings (Perla et al., 2021) and forecasting crime (Safat et al., 2021). However, these methods often act as black boxes, obfuscating errors and biases in their decision making. Interpretability methods attempt to explain how these models make their decisions. These explanations allow greater involvement of practitioners in the decision making process, a necessity for adoption in many contexts (Vellido, 2020).

Enguehard (2023) introduced ExtremalMask, a perturbation-based machine learning method for time series data that builds upon DynaMask (Crabbé & Van Der Schaar, 2021). In this paper, we investigate its reproducibility by validating the three primary claims posed in it. Beyond reproducing the original experiment, this work makes the following contributions:

- Explores the assumptions underlying the implementation of the 2 optimization problems in the original paper.

- Proposes an alternative saliency metric which takes into account the strength of the perturbations and analyzed its implications.

- Investigates whether the perturbations learned by ExtremalMask are realistic.

## 2 Scope of reproducibility

We test the three main claims found in Enguehard (2023):

**Claim 1** *On a synthetic dataset (HMM, Section 3.2.1), ExtremalMask best identified the non-salient features compared to 9 other methods, as measured by all relevant tested metrics specified in Section 3.4.1 besides AUP.*

---

[*]Equal contribution. Author ordering determined randomly.

**Claim 2** *On a real-life dataset (MIMIC-III, Section 3.2.3), ExtremalMask best identified the salient and non-salient features compared to 6 other methods, as measured by the relevant tested metrics specified in Section 3.4.1.*

**Claim 3** *ExtremalMask achieves better performance in the preservation game (explained in Section 3.1) compared to the deletion game on both datasets.*

To expand on the original paper, we first propose a more suitable saliency metric to the optimization problem solved and investigate its implications on ExtremalMask following extension:

**Extension 1** *We propose an alternative metric for measuring data saliency. Using ExtremalMask, we re-evaluate the information and entropy (defined in Section 3.4.1) of alternative saliency on HMM. Additionally, we perform a comparative study between identified salient data using the mask and alternative saliency metrics on MIMIC-III.*

Then we investigate the practicality of ExtremalMask with the following extension:

**Extension 2** *Using a synthetic dataset (HMM modified, Section 3.2.2), we explore the plausibility of the perturbations created with ExtremalMask by testing their probability relative to the original distribution.*

## 3 Methodology

Enguehard (2023) provided an open-source implementation of their proposed approach as part of the Python `tint` library[1]. The repository includes implementations of all methods used in this study, namely DynaMask (Crabbé & Van Der Schaar, 2021), Augmented Occlusion (Tonekaboni et al., 2020), DeepLift (Shrikumar et al., 2017), FIT (Tjoa & Guan, 2020), GradientShap (Lundberg & Lee, 2017), Integrated Gradients (Sundararajan et al., 2017), Lime (Ribeiro et al., 2016), Occlusion (Zeiler & Fergus, 2014), and Retain (Choi et al., 2016).

### 3.1 Model descriptions

ExtremalMask quantifies saliency of data in a multivariate time-series dataset for any predictive model $f : \mathbb{R}^{T \times D} \to \Omega$, in which $T$ is the time horizon, $D$ is the number of features and every prediction resides in the metric space $(\Omega, \mathcal{L})$. To identify such data, the model solves the following optimization problem, referred to as the **deletion** game:

$$\underset{\mathbf{M}, \boldsymbol{\Theta}}{\arg\min} \sum_{n=1}^{N} \lambda_1 |\mathbf{1} - \mathbf{M}_n| + \lambda_2 |\text{NN}(\mathbf{X}_n; \boldsymbol{\Theta}) - \mathbf{X_n}| - \mathcal{L}[f(\mathbf{X}_n), f(\Phi(\mathbf{X}_n, \mathbf{M}_n))], \tag{1}$$

where, $\mathbf{X} \in \mathbb{R}^{N \times T \times D}$ is the time-series dataset comprising of $N$ samples, $\mathbf{M} \in [0, 1]^{N \times T \times D}$ is the associated mask, $\text{NN} : \mathbb{R}^{T \times D} \to \mathbb{R}^{T \times D}$ is an arbitrary neural network (we refer to the network's output as **noise**), and $\Phi(\mathbf{X}_n, \mathbf{M}_n)$ is the perturbation function defined as:

$$\mathbf{M}_n \mathbf{X}_n + (1 - \mathbf{M}_n)\text{NN}(\mathbf{X}_n; \boldsymbol{\Theta}). \tag{2}$$

Intuitively, the deletion game rewards small perturbations that cause large changes in the output. Specifically, in Equation 1:

- The combination of $\lambda_1 |\mathbf{1} - \mathbf{M}_n|$ and $\lambda_2 |\text{NN}(\mathbf{X}_n; \boldsymbol{\Theta}) - \mathbf{X_n}|$ encourages the perturbed data $\Phi(\mathbf{X}_n, \mathbf{M}_n)$ to be close to the original data $\mathbf{X}_n$, incentivizing small perturbations.

- $-\mathcal{L}[f(\mathbf{X}_n), f(\Phi(\mathbf{X}_n, \mathbf{M}_n))]$ encourages the perturbed output $f(\Phi(\mathbf{X}_n, \mathbf{M}_n))$ to be far from the original output $f(\mathbf{X}_n)$, incentivizing large changes in the output after perturbation.

---

[1]https://github.com/josephenguehard/time_interpret

Note that the deletion game implemented in the original paper replaces:

- $\lambda_2|\text{NN}(\mathbf{X}_n; \boldsymbol{\Theta}) - \mathbf{X_n}|$ with $\lambda_2|\text{NN}(\mathbf{X}_n; \boldsymbol{\Theta})|$. By the triangle inequality, we have that $||\text{NN}(\mathbf{X}_n; \boldsymbol{\Theta}) - \mathbf{X_n}| - |\text{NN}(\mathbf{X}_n; \boldsymbol{\Theta})|| \leq |\mathbf{X_n}|$. Thus, ensuring in this modified optimization problem that, on average (across $n$), $\mathbf{X_n}$ is geometrically sufficiently close to $\mathbf{0}$ more accurately estimates the solution to the deletion game.

- $-\mathcal{L}[f(\mathbf{X}_n), f(\Phi(\mathbf{X}_n, \mathbf{M}_n))]$ with $\mathcal{L}[f(\mathbf{0}), f(\Phi(\mathbf{X}_n, \mathbf{M}_n))]$. Since, by the triangle inequality, it holds that $\mathcal{L}[f(\mathbf{0}), f(\mathbf{X}_n)] - \mathcal{L}[f(\mathbf{0}), f(\Phi(\mathbf{X}_n, \mathbf{M}_n))] \leq \mathcal{L}[f(\mathbf{X}_n), f(\Phi(\mathbf{X}_n, \mathbf{M}_n))]$, on average, $f(\mathbf{X}_n)$ should be geometrically sufficiently far away from $f(\mathbf{0})$ for it to be a good approximation. This change could also be motivated by the potential numerical problem associated with the negative distance being unbounded from below.

The **preservation** game, as the counterpart to the deletion game, is defined as:

$$\underset{\mathbf{M}, \boldsymbol{\Theta}}{\arg\min} \sum_{n=1}^{N} \lambda_1|\mathbf{M}_n| - \lambda_2|\text{NN}(\mathbf{X}_n; \boldsymbol{\Theta}) - \mathbf{X_n}| + \mathcal{L}[f(\mathbf{X}_n), f(\Phi(\mathbf{X}_n, \mathbf{M}_n))]. \tag{3}$$

In contrast to the deletion game, the preservation game rewards large perturbations that cause small changes in the prediction. To address the analogous numerical problem as in the deletion game, the original paper replaces $-\lambda_2|\text{NN}(\mathbf{X}_n; \boldsymbol{\Theta}) - \mathbf{X}_n|$ with $\lambda_2|\text{NN}(\mathbf{X}_n; \boldsymbol{\Theta})|$. Again, by triangle inequality, this solution again comes at the cost of the additional assumption that $\mathbf{X}_n$ on average being sufficiently far away from $\mathbf{0}$. Note that unlike the deletion game, the preservation game implemented in the original paper does not assume anything regarding $f$.

## 3.2 Datasets

The original paper utilized two datasets: a synthetic dataset generated by a Hidden Markov Model (HMM) and the MIMIC-III dataset, which both pose a classification problem. Only the training dataset is used to train the classifier $f$. ExtremalMask and other explainers (interpretability methods) aim to identify saliency of data for $f$'s predictions on the test dataset.

### 3.2.1 HMM (reproducibility study)

Enguehard (2023) used the implementation of the HMM dataset from Crabbé & Van Der Schaar (2021). However, we noticed some differences between the HMM dataset's description in the DynaMask paper and its implementation in `tint`. To foster reproducibility, we summarize the implemented dataset here, with the specific parameters described in Section A.1.1.

Let $1 \leq t \leq T$. Consider a 2-state HMM with hidden state at time $t$ as $s_t \in \{0, 1\}$. For a single sample $\mathbf{x} \in \mathbb{R}^{T \times D}$, the data at time $t$, $\mathbf{x}_t$, is sampled from a normal distribution with mean $\boldsymbol{\mu}_{s_t}$ and covariance matrix $\boldsymbol{\Sigma}_{s_t}$. Furthermore, the ground-truth label at time $t$, $\boldsymbol{y}_t$, is sampled from a Bernoulli distribution with probability:

$$p_t = \begin{cases} (1 + \exp(-2(\mathbf{x}_t)_2))^{-1} & (s_t = 0) \\ (1 + \exp(-2(\mathbf{x}_t)_3))^{-1} & (s_t = 1) \end{cases}.$$

We generate a dataset containing 1000 such samples with $T = 200$ and $D = 3$, i.e. $\mathbf{X} \in \mathbb{R}^{1000 \times 200 \times 3}$ and $\mathbf{Y} \in \mathbb{R}^{1000 \times 200}$. This dataset is split into 800 training samples and 200 test samples.

### 3.2.2 HMM (extension)

In Extension 2, we modify the HMM dataset introduced in Section 3.2.1 to facilitates the use of the forward algorithm (explained in Section A.2) to compute the probability of perturbed data occurrences within our HMM dataset to assess how likely the generated perturbations are. Specifically, we ensure that the Markovian

property is satisfied and reduce the time horizon to 50. Additionally, we rectify the asymmetry issue in the decay of transition probabilities present in the original HMM dataset. The modifications are detailed in Section A.1.1.

### 3.2.3 MIMIC-III

The MIMIC-III (Johnson et al., 2016) dataset includes vital sign information for over 40k intensive care unit patients at Beth Israel Deaconess Medical Center. From this dataset, we trained the classifier on 18,390 train samples and ExtremalMask on 4,598 test samples. Each sample contains a binary mortality outcome (sampled patients had a 9% mortality rate) and 31 vital signs measured over 48 hour-long time steps. In order to replace missing values, we use the previous values when possible and a standard value otherwise.

### 3.3 Hyperparameters

We use the default hyperparameters provided by the `tint` library. The classifier and the NN within ExtremalMask both utilize a GRU (Cho et al., 2014) architecture across all our experiments. These choices are consistent with the original paper.

### 3.4 Experimental setup and code

#### 3.4.1 Metrics

Suppose there exists an index set $\mathbf{A}$ of $\mathbf{X} \in \mathbb{R}^{N \times T \times D}$ such that $\mathbf{A}_n$ is the index set of the ground truth non-salient features for $\mathbf{X}_n$. To assess the minimal impact of the identified non-salient data on predictions, Enguehard (2023) employs the following four metrics when perturbing the non-salient data:

**Area Under Precision (AUP) and Area Under Recall (AUR)** These metrics gauge the similarity between predictions on perturbed and original data. A higher value indicates better performance, signifying perturbing non-salient features marginally impacts the predicted classes.

**Information ($I$)** Defined as $I_{\mathbf{M}}(A) = -\sum_{n=1}^{N} \sum_{(t,d) \in A} \log(1 - \mathbf{M}_{ntd})$, $I$ is higher when the perturbed data on average places more weight on $\mathbf{X}$. A higher information is desired, signifying a more informative mask.

**Entropy ($E$)** Defined as $E_{\mathbf{M}}(A) = \sum_{n=1}^{N} \sum_{(t,d) \in A} \mathbf{M}_{ntd} \log(\mathbf{M}_{ntd}) + (1 - \mathbf{M}_{ntd}) \log(1 - \mathbf{M}_{ntd})$. Entropy is low for masks with extreme values and high for masks with values around 0.5. Therefore, a lower entropy is desired, signifying the learned mask being more confident in the identified salient or non-salient data.

For a real-life dataset like MIMIC-III, the same metrics cannot be used as the ground-truth salient data is unknown. Instead, Enguehard (2023) considers the 20% of the data with the highest mask values to be the identified salient data, where we refer to the 20% as the saliency ratio henceforth, and introduces the following metrics:

**Accuracy (Acc)** Measures whether perturbing identified salient data results in an accurate prediction. A lower value is preferred, signifying more important features being perturbed.

**Cross-Entropy (CE)** Measures change in predicted probability for the correct class. A higher value is preferred, suggesting perturbed data is salient as it decreases the probability of the original prediction.

**Comprehensiveness (Comp) and Sufficiency (Suff)** Both metrics are defined as the average distance between the probability of the correct class on original and perturbed data. However, when calculating sufficiency, the 80% of data with the lowest mask values (identified non-salient data) are perturbed instead. A higher value is desired, signifying that perturbing salient data influences the probability of predicting the correct class.

Perturbing MIMIC-III data refers to replacing a feature in a data-point with the feature's average over the 48 hours sample unless otherwise specified.

### 3.4.2 Reproducibility study

The repository from the original paper includes code for training the classifier and explainers as well as code for outputting results. To facilitate comparisons with the original paper, we made minimal edits to the original code. These edits include code for saving and loading checkpoints of neural networks.

In all our experiments, we seed PyTorch and other libraries to improve experimental reproducibility. The authors mentioned in our communication that they opted not to do this in their paper to reduce experiment runtime. In an additional change, unfortunately, for Claim 2, we chose not to retrieve the results for certain computationally intensive methods. To enable others to reproduce our results, we provide job scripts for executing our experiments on a computer cluster, and shell scripts for running the code locally.

### 3.4.3 Additional experiments

In this section, we describe the approaches of tackling each of the extensions introduced in Section 2.

**Extension 1** The original saliency metric considers only the mask. However, the mask alone fails to effectively represent the magnitude of the difference between the perturbed and original data. Specifically, both $\text{NN}(\mathbf{X}_n)_{td}$ being arbitrarily close to or being distant from $\mathbf{X}_{ntd} \in \mathbb{R}$ could result in low values of the mask in the preservation game. To tackle this, we propose the following saliency metric $\mathbf{S}$ given $\mathbf{X}$:

$$\mathbf{S}_{ntd} \stackrel{\text{def}}{=} 1 - \frac{|\mathbf{X}_{ntd} - \Phi(\mathbf{X}_n, \mathbf{M}_n)_{td}|}{\|\mathbf{X}_n - \Phi(\mathbf{X}_n, \mathbf{M}_n)\|_\infty}. \tag{4}$$

A higher value indicates a smaller perturbation, identifying salient data in the preservation game. Note that compared to the mask, we would always expect alternative saliency to be at least as informative about the perturbation size, and therefore also at least as informative about the learned saliency of the data by ExtremalMask. We calculate the information and entropy of this metric to evaluate the informativeness of the perturbations learned by ExtremalMask.

Furthermore, we conduct a comparative analysis between the salient data identified by the two saliency metrics in the preservation game. We examine differences on a local and global scale. Since we are interested in the data that led to the classifier to correctly classify a patient as dead, we exclusively consider accurately classified dead patients within MIMIC-III in this study.

For a local analysis, we pick an arbitrary correctly classified dead patient in MIMIC-III and plot the saliency results. In one plot, we classified data points as salient if they were predicted to have a saliency value in the top 20% in its sample. This choice of percentage for hard classification of salient points is somewhat arbitrary but aids us in visually assessing data with multiple features and time steps. When employing these visualizations, we recommend exploring different top percentages and recognize that the ideal percentage might vary depending on dataset size. Then, we contrast the heatmap based on the mask and alternative saliency values for that patient. We refer to these heatmaps as saliency maps.

For a global analysis, we calculate the weighted averages of the mask and the alternative saliency for every feature over time and across all patients in the MIMIC-III test dataset. Enguehard (2023) found that data from later timesteps (closer to the patient's death) had greater saliency than data from earlier periods. To ascertain which features carry more saliency in these later time steps, we compute three variants of weighted averages with the following weights applied to saliency values for data at time step $t$ ($1 \leq t \leq T$):

1. No decay: $1$.

2. Linear decay: $1 - \frac{T-t}{T}$.

3. Exponential decay: $e^{-\frac{T-t}{T}}$.

Moreover, we employ Monte Carlo's 95% confidence intervals for the attributions.

**Extension 2** Perturbed data generated through perturbation methods may extend beyond their original distribution, becoming out of distribution (OOD). The classifier's decisions on these unrealistic perturbations are less meaningful, since these perturbations are unlikely to occur during the classifier's training process or its deployment. As a result, we wanted to test if ExtremalMask produced less realistic, and therefore lower quality, perturbations.

Specifically, we conducted an experiment for ExtremalMask on the modified HMM dataset (Section 3.2.2) with the test dataset $\mathbf{X} \in \mathbb{R}^{200 \times 50 \times 3}$. Consider the stochastic process $Z = (Z_t)_{t=1}^{50}$ such that $Z_t \sim N(\boldsymbol{\mu}_{s_t}, \boldsymbol{\Sigma}_{s_t})$. We classify perturbed data $\mathbf{X}_n' \in \mathbb{R}^{50 \times 3}$, derived from the raw data $\mathbf{X}_n$, as OOD when:

$$\log f_Z(\mathbf{X}_n') < \mathrm{median}((\log f_Z(\mathbf{X}_n))_{n=1}^N) - \lambda \cdot \mathrm{IQR}((\log f_Z(\mathbf{X}_n))_{n=1}^N), \tag{5}$$

where, $f_Z$ is the probability density function of $Z$ and $\lambda \geq 0$ controls the tolerance for identifying OOD instances. This approach follows the same principle as using $Z$-scores to assess the probability of a datapoint belonging to a normal distribution. However, instead of relying on the conventional mean and standard deviation, we use the more robust median and interquartile range (IQR) measures. Using the forward algorithm (Section A.2) that leverages the Markovian assumption to confine the search space, we could evaluate $f_Z(x)$ for any $x \in \mathbb{R}^{50 \times 3}$. To prevent numerical underflow, we use a shortened time horizon of 50 timesteps.

In our codebase, we provide Jupyter notebooks that enable reproduction of our additional experiments.

## 3.5 Computational requirements

We used 1 NVIDIA A100 GPU and nine CPUs on a computer cluster for all of our reproducibility experiments. Our experiments took a total of around 81 hours to run, with the time used per claim specified in Section A.3. We ran all of our extensions on CPU (Intel i7-4720HQ), taking negligible time.

## 4 Results

Our reproduction study confirmed all three claims laid out in Section 2, with the exception of the AUR result in Claim 1 and Claim 3 on the HMM dataset. Specifically, ExtremalMask outperforms the other methods on all metrics for the MIMIC-III dataset, and has strong performance on the HMM dataset, with notably strong values for information and entropy. Note that as the results presented in Enguehard (2023) were obtained without seeding, drawing comparative conclusions with our seeded results poses challenges. Our extensions provide a deeper empirical intuition into the perturbation methods and further confidence why these methods are reasonable for explaining ML models.

Table 1: Our results reported on the HMM dataset as average $\pm$ standard deviation over 5 folds. Columns labeled with the suffix -R denote diff-to-std ratios, obtained by dividing the difference between our and the original paper's results by the standard deviation from the original paper. Differences falling outside 2 standard deviations are underlined.

| Method | AUP ↑ | AUP-R | AUR ↑ | AUR-R | $I$ ↑ | $I$-R | $E$ ↓ | $E$-R |
|---|---|---|---|---|---|---|---|---|
| Aug. Occlusion | $0.867 \pm 0.006$ | 2.60 | $0.351 \pm 0.006$ | 1.48 | $3.03\mathrm{E}{+}4 \pm 601$ | 1.65 | $3.40\mathrm{E}{+}4 \pm 232$ | 0.877 |
| DeepLift | $\mathbf{0.931} \pm 0.003$ | 0.579 | $0.328 \pm 0.009$ | 11.5 | $2.98\mathrm{E}{+}4 \pm 890$ | 1.45 | $3.00\mathrm{E}{+}4 \pm 286$ | 0.879 |
| DynaMask (MSE) | $0.376 \pm 0.003$ | 16.7 | $\mathbf{0.771} \pm 0.002$ | 0.308 | $1.05\mathrm{E}{+}5 \pm 346$ | 2.60 | $2.53\mathrm{E}{+}4 \pm 59.2$ | 2.34 |
| DynaMask (CE) | $0.341 \pm 0.003$ | 18.5 | $0.569 \pm 0.002$ | 7.46 | $1.23\mathrm{E}{+}5 \pm 1.22\mathrm{E}{+}3$ | 6.77 | $9.32\mathrm{E}{+}3 \pm 161$ | 39.3 |
| Fit | $0.474 \pm 0.034$ | 4.08 | $0.576 \pm 0.067$ | 1.59 | $7.72\mathrm{E}{+}4 \pm 8.51\mathrm{E}{+}3$ | 0.117 | $3.30\mathrm{E}{+}4 \pm 1.62\mathrm{E}{+}3$ | 0.946 |
| GradientShap | $0.886 \pm 0.004$ | 1.23 | $0.283 \pm 0.007$ | 8.73 | $2.55\mathrm{E}{+}4 \pm 697$ | 1.06 | $2.78\mathrm{E}{+}4 \pm 282$ | 0.806 |
| IG | $\mathbf{0.931} \pm 0.003$ | 0.684 | $0.323 \pm 0.008$ | 11.9 | $2.92\mathrm{E}{+}4 \pm 817$ | 1.40 | $2.99\mathrm{E}{+}4 \pm 293$ | 0.919 |
| Lime | $0.950 \pm 0.003$ | 1.06 | $0.301 \pm 0.007$ | 17.1 | $2.73\mathrm{E}{+}4 \pm 638$ | 1.16 | $2.84\mathrm{E}{+}4 \pm 259$ | 0.430 |
| Occlusion | $0.919 \pm 0.003$ | 1.66 | $0.283 \pm 0.005$ | 18.3 | $2.56\mathrm{E}{+}4 \pm 530$ | 1.54 | $2.77\mathrm{E}{+}4 \pm 212$ | 1.48 |
| Retain | $0.681 \pm 0.113$ | 0.409 | $0.280 \pm 0.049$ | 4.15 | $2.23\mathrm{E}{+}4 \pm 4.85\mathrm{E}{+}3$ | 4.33 | $2.95\mathrm{E}{+}4 \pm 2.67\mathrm{E}{+}3$ | 3.45 |
| ExtremalMask (MSE) | $0.904 \pm 0.010$ | 0.633 | $0.760 \pm 0.004$ | 1.62 | $\mathbf{2.93\mathrm{E}{+}5} \pm 4.51\mathrm{E}{+}3$ | 1.07 | $\mathbf{7.51\mathrm{E}{+}3} \pm 183$ | 0.927 |
| ExtremalMask (CE) | $0.913 \pm 0.009$ | 0.933 | $0.666 \pm 0.012$ | 8.85 | $1.54\mathrm{E}{+}5 \pm 6.58\mathrm{E}{+}3$ | 10.4 | $1.59\mathrm{E}{+}4 \pm 436$ | 16.7 |

## 4.1 Claim 1 - ExtremalMask best (besides in AUP) identifies non-salient features on HMM

Unlike all other methods, only DynaMask and ExtremalMask used the mean squared error (MSE) loss function on HMM in the provided code. Notably, those two methods used cross-entropy (CE) loss on MIMIC-III. Therefore, to facilitate fair comparison with other methods, we also test the results of these two methods trained on HMM with the cross-entropy (CE) loss.

Table 1 corroborates that ExtremalMask (MSE) outperformed other methods in information and entropy; whereas the original paper found ExtremalMask had the best performance in AUR, we found that it had the second best performance for both the MSE and CE variants. Thus, our reproducibility results only partially support Claim 1. In addition, we found that ExtremalMask (MSE) outperforms ExtremalMask (CE) in both information and entropy. Nevertheless, with ExtremalMask (CE), it still had the best performance in information and entropy compared to the other methods.

In addition, Table 1 shows that our calculated information and entropy differed by two orders of magnitude from the original paper. We have reported this discrepancy to the authors, who indicated that their paper included a mistake and sent us the updated values for these metrics.

Using these updates values, in Table 1, we present the absolute difference between our results and those of the authors, divided by the standard deviation reported in the original paper. We refer to these as the diff-to-std ratios henceforth. For AUR and AUP, we used the standard deviations reported in the original paper, whereas we use the updated standard deviations for information and entropy updated by the authors. Many of our results do not lie within 2 times the standard deviation. The significant magnitudes of some of the diff-to-std ratios, such as that of DynaMask, suggest that randomness is an unlikely cause. The diff-to-std ratios of ExtremalMask (CE) seem to suggest that the values in the original paper are obtained by training ExtremalMask using MSE.

## 4.2 Claim 2 - ExtremalMask best identifies salient and non-salient features on MIMIC-3

In Table 2, we observe that ExtremalMask outperformed all other tested methods on the MIMIC-III dataset across all four relevant metrics, thereby substantiating Claim 2. In addition, Section A.5 shows that this result held across variations of saliency ratios of 10%, 30% 40%, and 60%, as well as, replacing the identified salient data with **0** instead of its average. The diff-to-std ratios in Table 2 once again highlight a significant deviation of our DynaMask result from those reported in the original paper, when compared to other methods.

Table 2: Our results reported on the MIMIC-III dataset as average $\pm$ standard deviation over 5 folds. Columns labeled with the suffix -R denote diff-to-std ratios, obtained by dividing the difference between our and the original paper's results by the standard deviation from the original paper. Differences falling outside 2 standard deviations are underlined.

| Method | Acc ↓ | Acc-R | Comp ↑ | Comp-R | CE ↑ | CE-R | Suff ↓ | Suff-R |
|---|---|---|---|---|---|---|---|---|
| Aug. Occlusion | 0.987 $\pm$ 0.002 | 2.00 | 1.68E-03 $\pm$ 7.21E-04 | 1.17 | 9.83E-02 $\pm$ 6.75E-03 | 0.105 | 2.27E-03 $\pm$ 2.12E-03 | 0.201 |
| DeepLift | 0.987 $\pm$ 0.002 | 0.500 | 8.98E-04 $\pm$ 1.65E-03 | 1.11 | 9.74E-02 $\pm$ 7.15E-03 | 0.106 | 3.21E-03 $\pm$ 7.88E-04 | 0.302 |
| DynaMask | 0.990 $\pm$ 0.002 | 0 | 4.31E-03 $\pm$ 9.52E-04 | 5.53 | 0.1 $\pm$ 0.006 | 0.697 | 5.40E-04 $\pm$ 1.48E-03 | 1.74 |
| IG | 0.986 $\pm$ 0.003 | 0.667 | 1.50E-03 $\pm$ 1.92E-03 | 0.851 | 9.84E-02 $\pm$ 7.19E-03 | 0.13 | 2.62E-03 $\pm$ 5.42E-04 | 0.339 |
| Occlusion | 0.987 $\pm$ 0.002 | 1.00 | -3.60E-04 $\pm$ 9.90E-04 | 2.02 | 9.62E-02 $\pm$ 6.73E-03 | 0.194 | 4.58E-03 $\pm$ 1.68E-03 | 7.52E-03 |
| Retain | 0.987 $\pm$ 0.003 | 2.00 | -2.08E-03 $\pm$ 1.57E-03 | 1.28 | 9.49E-02 $\pm$ 7.33E-03 | 0.337 | 7.61E-03 $\pm$ 8.94E-04 | 9.91E-02 |
| ExtremalMask | **0.983** $\pm$ 0.003 | 0.633 | **1.23E-02** $\pm$ 2.16E-03 | 1.62 | **0.114** $\pm$ 0.009 | 1.07 | **-8.07E-03** $\pm$ 2.21E-03 | 0.927 |

Table 3: Our results reported on the HMM dataset for the two optimization problems as average $\pm$ standard deviation over 5 folds. Columns labeled with the suffix -R denote diff-to-std ratios, obtained by dividing the difference between our and the original paper's results by the standard deviation from the original paper. Differences falling outside 2 standard deviations are underlined.

| Game | AUP ↓ | AUP-R | AUR ↓ | AUR-R | $I$ ↓ | $I$-R | $E$ ↓ | $E$-R |
|---|---|---|---|---|---|---|---|---|
| Preservation | **0.904** $\pm$ 0.010 | 0.633 | 0.760 $\pm$ 0.004 | 1.62 | **2.93E+5** $\pm$ 4.51E+3 | 1.07 | **7.51E+3** $\pm$ 183 | 0.927 |
| Deletion | 0.341 $\pm$ 0.001 | 1.67 | **0.898** $\pm$ 0.002 | 2.92 | 2.52E+5 $\pm$ 1.56E+3 | 0.781 | 1.21E+4 $\pm$ 191 | 0.220 |

Table 4: Our results reported on the MIMIC-III dataset for the two optimization problems as average ± standard deviation over 5 folds. Columns labeled with the suffix -R denote diff-to-std ratios, obtained by dividing the difference between our and the original paper's results by the standard deviation from the original paper. Differences falling outside 2 standard deviations are underlined.

| Game | Acc ↓ | Acc-R | Comp ↑ | Comp-R | CE ↑ | CE-R | Suff ↓ | Suff-R |
|---|---|---|---|---|---|---|---|---|
| Preservation | **0.984** ± 0.003 | 0.75 | **1.14e-02** ± 0.001 | 1 | **0.11** ± 0.006 | 0.875 | **-8.38e-03** ± 0.001 | 1 |
| Deletion | 0.996 ± 0.001 | 2.5 | -0.003 ± 0.001 | 0.5 | 0.088 ± 0.003 | 0.455 | 0.010 ± 0.004 | 0.5 |

Table 5: Information and entropy of the two saliency metrics over 5 folds.

| Saliency metric | $I$ ↑ | $E$ ↓ |
|---|---|---|
| **M** | 2.93E+5 ± 4.51E+3 | 7.51E+3 ± 1.83E+2 |
| **S** | **3.33E+5** ± 2.99E+3 | **7.29E+3** ± 1.27E+2 |

### 4.3  Claim 3 - **ExtremalMask performs better in preservation than deletion game on both datasets**

Table 4 illustrates that the preservation game outperforms the deletion game across all metrics except for AUR on HMM. On the other hand, the preservation game outperforms the deletion game across all metrics on MIMIC-III. A possible explanation for the preservation game's superiority lies in its avoidance of additional assumptions regarding the classifier, as highlighted in Section 3.1. Concluding which assumption regarding the data holds true for the two methods requires a rigorous definition of what it means for the data to be sufficiently close or far away from **0**. Formalizing these assumptions would help determine which implemented optimization problem is more suitable for a specific dataset, supplementing our less-generalizable empirical study.

### 4.4  Extension 1 - **Comparative study between 2 saliency metrics**

In Table 5, we observe that the perturbations learned by ExtremalMask, as quantified by the alternative saliency metric (**S**), is more informative for identifying the salient data compared to its proxy, i.e. the mask (**M**).

Figure 1 visualizes the saliency maps and the salient data generated using the two saliency metrics for an arbitrarily-selected correctly-classified dead patient, specifically patient 64 from fold 0. Our visual analysis yielded the following conclusions:

1. The alternative saliency values tend to be significantly higher and less polarized than the mask values. This suggests that the noise learned (output of NN) tends to be close to the original data, resulting in limited perturbations.

2. The alternative saliency values tend to be less sporadic than the mask values. This suggests that features identified as salient at one point in time persist in their saliency over an extended period with alternative saliency. This stability in salient features over time allows for simpler and more interpretable explanations.

Next, we performed global analysis to identify the most salient features. Figure 2 shows substantial differences in magnitudes among attributions calculated using different saliency metrics. One would expect that lower mask values lead to larger perturbations and lower alternative saliency values. In turn, this would result in the mask being similar in value as alternative saliency. The observed discrepancy, however, is justifiable when the noise (output of NN) learned by ExtremalMask is, on average, close to the original data, limiting the impact of the mask on the size of the perturbations.

Moreover, we note that the confidence intervals for the mask values per feature are larger than those for alternative saliency. This indicates that the mask values tend to be more polarized over time compared to alternative saliency, which aligns with our analysis for the saliency maps of patient 64.

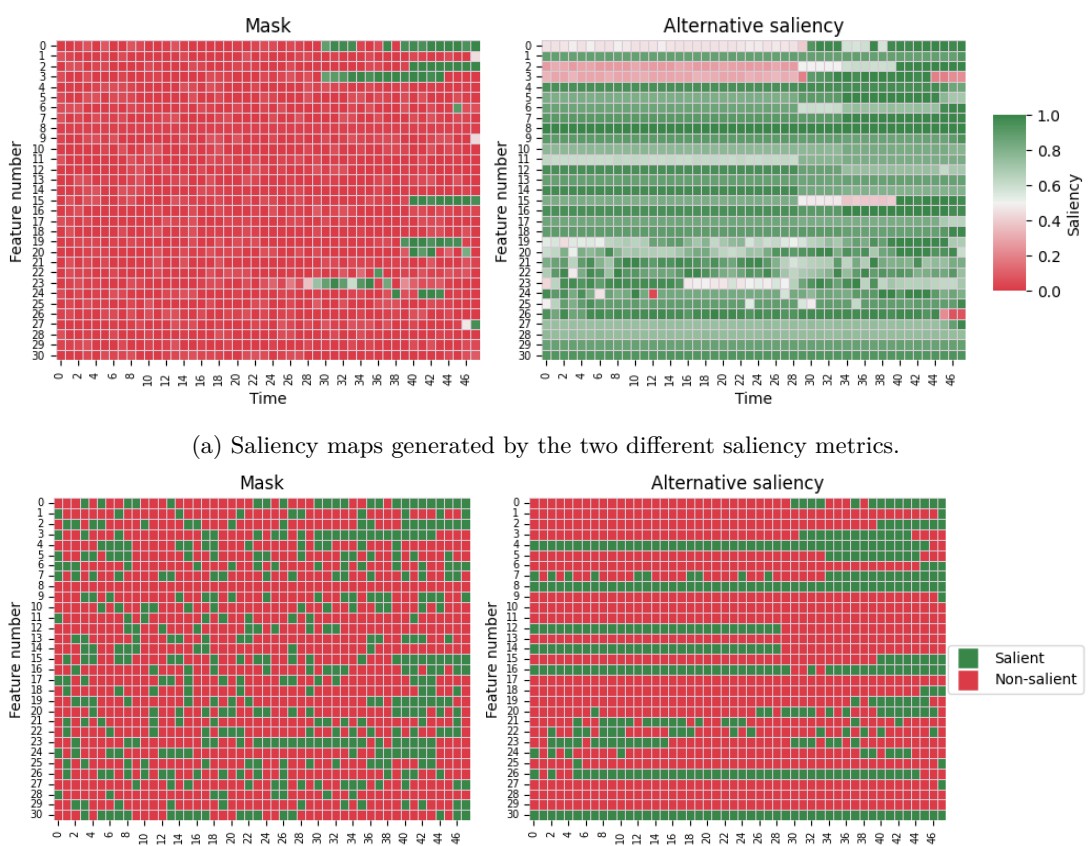

(a) Saliency maps generated by the two different saliency metrics.

(b) Binary saliency maps where only data with top 20% highest saliency values are considered as salient.

Figure 1: Saliency maps of two saliency metrics for a correctly classified dead patient (patient 64, fold 0). These maps show saliency of different features at different times for predicting the patient's mortality.

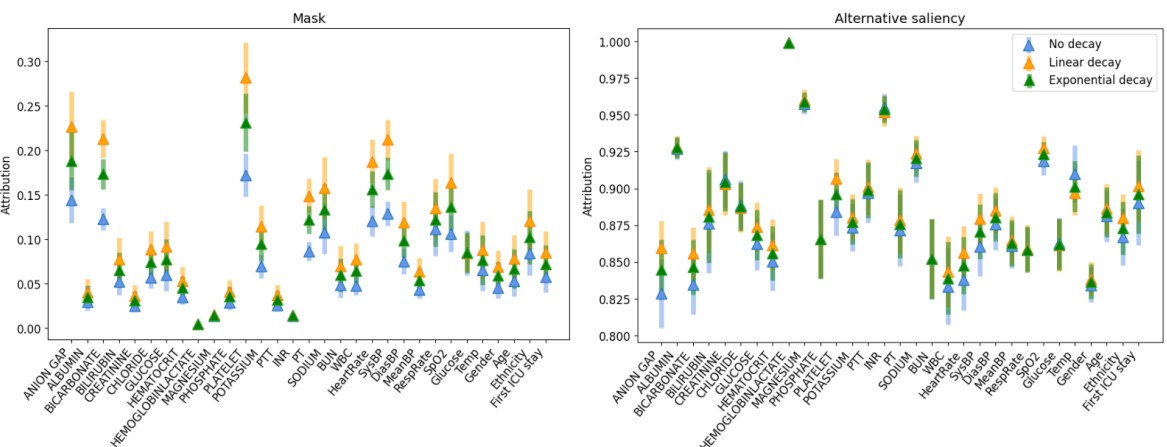

Figure 2: Attribution for a feature is the weighted averages of its saliency over time across all patients in the MIMIC-III test dataset, where the weights correspond to different form of decays introduced in Section 3.2.1. The error bars are the 95% confidence intervals of the Student's t-distribution. A higher value of attribution implies an identified greater importance for the classifier's prediction.

We found that some features seemed to be sensitive to the decay used. Certain features had higher attribution when using exponential or linear decay relative to no decay, indicating that they grew in salience over time. Other features were relatively invariant - their attributions overlapped using 3 forms of decays. These findings may enable practitioners to learn from or to debug the classifier. For example, we found that temperature became less relevant over time. This conflicted with our expectation that temperature measurements later in the time series (occurring closer to death) would be more important. Equally surprisingly, we found that gender became more salient over time, whereas we had expected gender salience to be time-invariant. These findings may reflect an error in either the classifier or in naively interpreting the results of ExtremalMask.

### 4.5 Extension 2 - Are the perturbed data of ExtremalMask realistic?

Finally, we found that the samples perturbed by ExtremalMask tend to be within distribution. Figure 3 shows that most of the perturbed samples are more probable than their original unperturbed counterparts. In fact, we observe that for any $\lambda \geq 0$ in Equation 4, every perturbed data point is not considered as OOD.

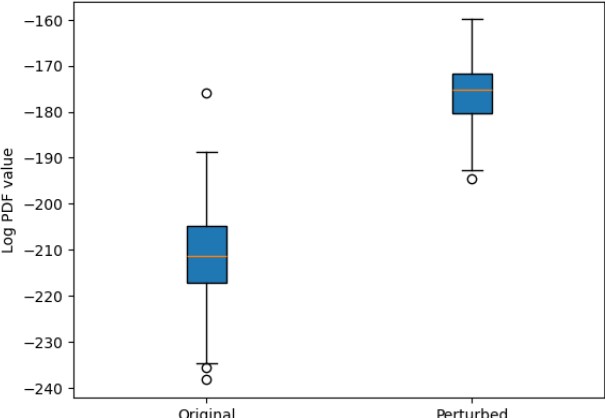

Figure 3: Boxplot of the log-probability of the original and perturbed test data of the modified HMM. The higher the value, the more probable the datapoint.

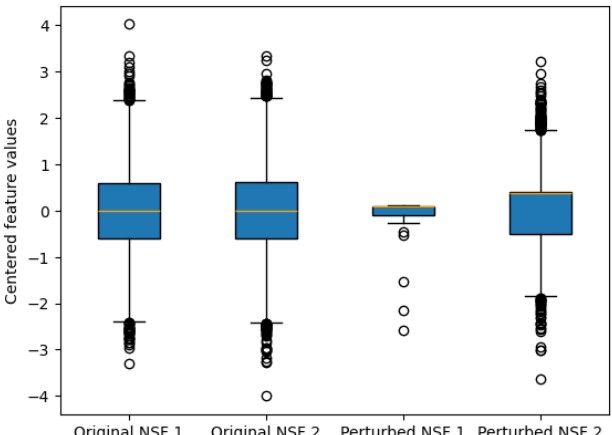

Figure 4: Boxplot of the difference between non-salient features and the mean of the feature's normal distribution for the original and perturbed test data of the modified HMM dataset. NSF stands for non-salient feature, where NSF 1 is the first feature (which remains non-salient throughout time) and NSF 2 is the combination of non-salient parts of feature 2 and 3 (see Section A.1.1 for ground-truth label generation).

Figure 4 provides an explanation for why the perturbed data became more probable. Feature 1 of the perturbed data is clustered noticeably closer to the mean of its distribution, as evidenced by the small width of the feature's zero-centered IQR. On the other hand, we had anticipated that ExtremalMask would identify the non-salient parts of features 2 and 3, and that, after perturbation, these values would have a similarly narrow IQR. However, the IQR of these parts appear to be relatively large in the figure, suggesting that ExtremalMask struggled to identify which parts of these features were non-salient.

In summary, ExtremalMask does not seem to generate OOD perturbations on the modified HMM dataset, thereby reinforcing the credibility of the associated salient data identified. This implies that ExtremalMask may generate comparable realistic perturbed data on analogous, relatively straightforward distributions.

## 5 Discussion

This paper presents a reproducibility study of Enguehard (2023) and our findings generally support the conclusions of the original paper. With our extensions, we (1) proposed a new saliency metric and offered an analysis on the relation between noise and the original data on MIMIC-III, and (2) found out that perturbed data learned by ExtremalMask on the modified HMM dataset is more probable than the original data.

We would suggest the following areas for future research:

- Formalizing sufficiently close or far away that are present as assumptions for the preservation and deletion game introduced in Enguehard (2023).

- Domain experts could be consulted to validate whether the generated explanations appropriately identify salient features. For instance, considering the top-k identified salient features from Figure 2, would a doctor also agree that these features are most important for determining patient mortality?

- It seems to be possible to get rid of the mask and instead just add the noise to the data as the expressiveness of ExtremalMask with this modification would remain unchanged. In this case, the only natural saliency metric would be alternative saliency.

### 5.1 What was easy

Overall, the authors maintained a very well-structured and well-modularized code base with good documentation. They maintain this code base as a publicly available resource for time series experiments, a testament to their dedication to reproducibility. To run the HMM experiment, we simply had to follow the documentation to create a conda environment (using the provided `.yml` file) and run the provided experiment shell script. It was also easy to imagine extensions because the original paper had a relatively straightforward premise with deep implications.

### 5.2 What was difficult

We spent some days debugging the difference between our results for entropy and information and the incorrect values reported in the original paper. We also had some difficulty loading check-pointed models due to differences between CUDA and CPU devices as well as the relatively old PyTorch version used in the `tint` library's environment.

Finally, like most new methods, the theoretical basis of this method is not fully developed. This made it difficult to build upon previous theory in order to create a rigorous analysis.

### 5.3 Communication with original authors

We contacted the authors twice by email, who gave us swift and thorough feedback to several lengthy theoretical and reproducibility questions. The authors have also briefly reviewed this manuscript before our submission and did not point out major issues.

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

# A   Appendix

## A.1   HMM dataset

In this section, we describe the specific parameters used for the two HMM datasets described in Section A.1.1 and Section A.1.2.

### A.1.1   Reproducibility study

Recall the 2-state HMM with hidden state at time t is $s_t \in \{0, 1\}$ as described in Section 3.2.1. The associated initial distribution vector $\boldsymbol{\pi}$ and transition matrix at time $t$, $\boldsymbol{\Sigma}_t$, are defined as

$$\boldsymbol{\pi} = (0.5, 0.5) \qquad \boldsymbol{\Sigma}_t = \begin{bmatrix} 0.95 & 0.05 \\ 0.05 + \frac{\theta((s_i)_{i=1}^t)}{500} & 0.95 - \frac{\theta((s_i)_{i=1}^t)}{500} \end{bmatrix},$$

where, for $1 \le m \le t$, $\theta((s_i)_{i=1}^t) = m$ if and only if $s_i = 1$ for all $t - m \le i < t$ and $s_{t-m-1} = 0$. To generate a single time-series sample $\mathbf{x} \in \mathbb{R}^{T \times D}$, we have $\mathbf{x}_t \sim \mathcal{N}(\boldsymbol{\mu}_{s_t}, \boldsymbol{\Sigma}_{s_t})$ with

$$\boldsymbol{\mu}_0 = \begin{bmatrix} 0.1 \\ 1.6 \\ 0.5 \end{bmatrix}, \boldsymbol{\mu}_1 = \begin{bmatrix} -0.1 \\ -0.4 \\ -1.5 \end{bmatrix}, \boldsymbol{\Sigma}_0 = \begin{bmatrix} 0.801 & 0 & 0 \\ 0 & 0.801 & 0.01 \\ 0 & 0.01 & 0.801 \end{bmatrix}, \boldsymbol{\Sigma}_1 = \begin{bmatrix} 0.801 & 0.01 & 0 \\ 0.01 & 0.801 & 0 \\ 0 & 0 & 0.801 \end{bmatrix}.$$

### A.1.2   Additional details for Section 3.2.2

The modified HMM differs from the original in the following properties:

1. Shortened sequence length, T=50

2. New hidden state transition probabilities:

$$\boldsymbol{\Sigma}_t = \begin{bmatrix} 0.95 - p_t & 0.05 + p_t \\ 0.05 + p_t & 0.95 - p_t \end{bmatrix}$$

where

$$p_t = t/500$$

## A.2   The Forward algorithm (FA)

FA is a dynamical programming algorithm used to calculate the marginal probability of a sequence of labels generated by a HMM. This probability is marginal in the sense that it's calculated by integrating over all possible hidden state sequences. FA reduces the time complexity of calculating the marginal label probability from exponential (in hidden state sequence length) to linear.

Let $N$ be the number of different hidden states, $K$ the number of labels, and $M$ the sequence length of a HMM. Furthermore let $(L_k)_{k=1}^M$ and $(H_k)_{k=1}^M$ be the random variables representing the labels and hidden states respectively. Let $(l_k)_{k=1}^M$ be the observed label sequence. We assume that the hidden states take values in $\{1, .., N\}$. Let $p$ represent the joint distribution of all hidden states and labels.

The main idea of the FA is the definition:

$$\alpha_{i,j} \overset{\text{def}}{=} p(L_1 = l_1, L_2 = l_2, ..., L_i = l_i, H_i = j) \tag{6}$$

And the following set of equations:

$$\alpha_{1,j} = \pi_j p(L_1 = l_1 | H_1 = j) \tag{7}$$

$$\alpha_{i,j} = \sum_{k=1}^N \alpha_{i-1,k} p(H_i = j | H_{i-1} = k) p(L_i = l_i | H_i = j) \tag{8}$$

Where $\pi_j$ are the initial hidden state probabilities. The second and third term inside the sum in equation 8 are called the transition and emission probabilities. The FA consists of filling the $M \times N$ matrix $\alpha$ row by row. The final result

$$p(L_1 = l_1, L_2 = l_2, ..., L_M = l_M) \tag{9}$$

is given by the sum of its last row.

### A.3   Time used per claim

|         | Hours |
|---------|-------|
| Claim 1 | 40    |
| Claim 2 | 40    |
| Claim 3 | 1     |

Table 6: Approximated number of hours used per claim on Snellius supercluster

### A.4   Weights for different decays in Section 3.4.3

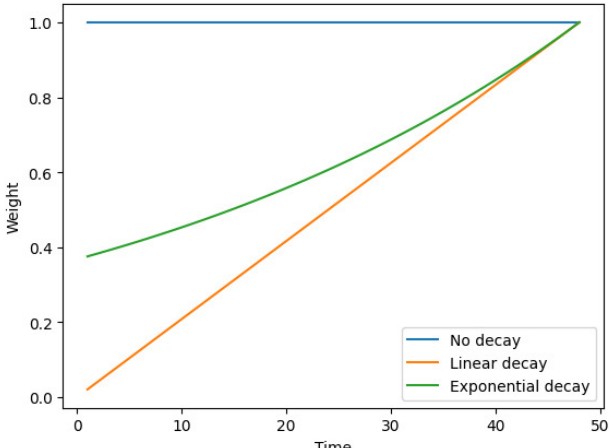

Figure 5: Weights of the different forms of decays as described in Section 3.4.3 with time horizon 48.

## A.5   Complete metrics of MIMIC-III experiment

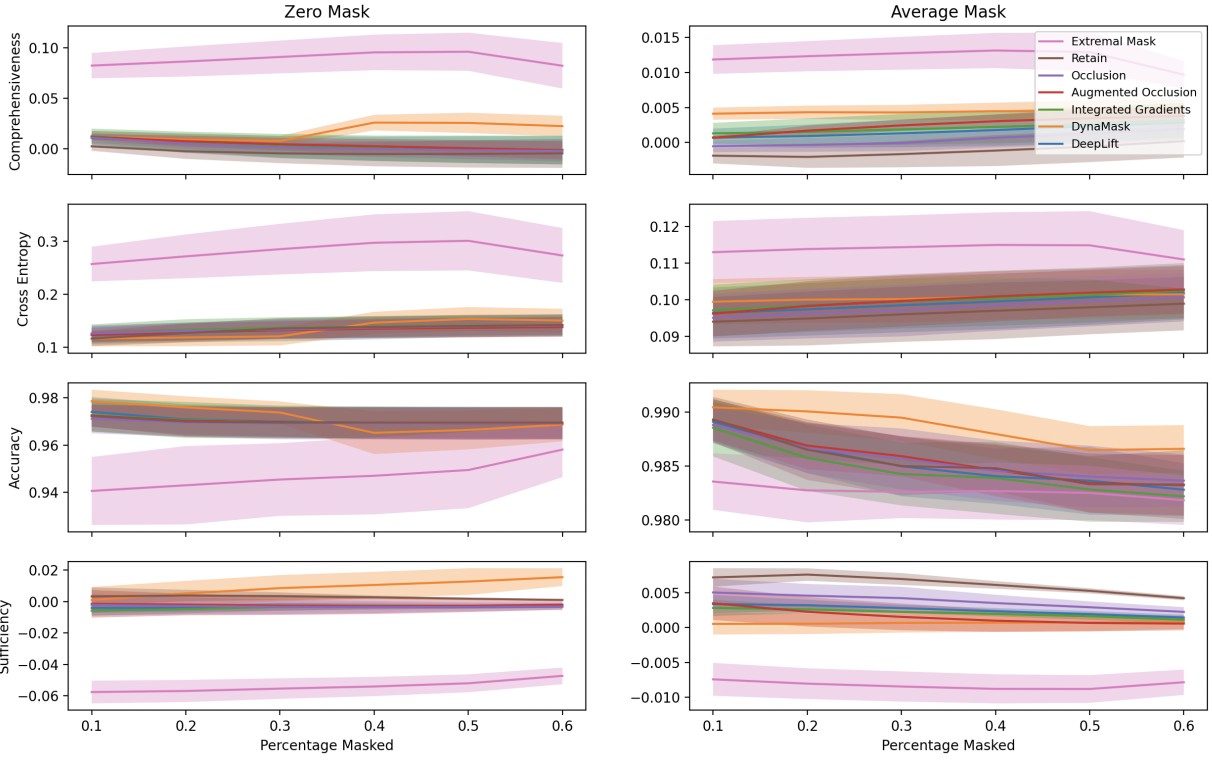

Figure 6: Results on MIMIC-III using different variations of the experiment. Each row corresponds to a different metric. For the left column, we masked the top x% of values using zero as a mask. For the right column, we used the feature's sample average. In all variations, ExtremalMask had the best performance (lowest accuracy and sufficiency, highest comprehensiveness and cross entropy).

## A.6 Additional visualizations of MIMIC-III experiment

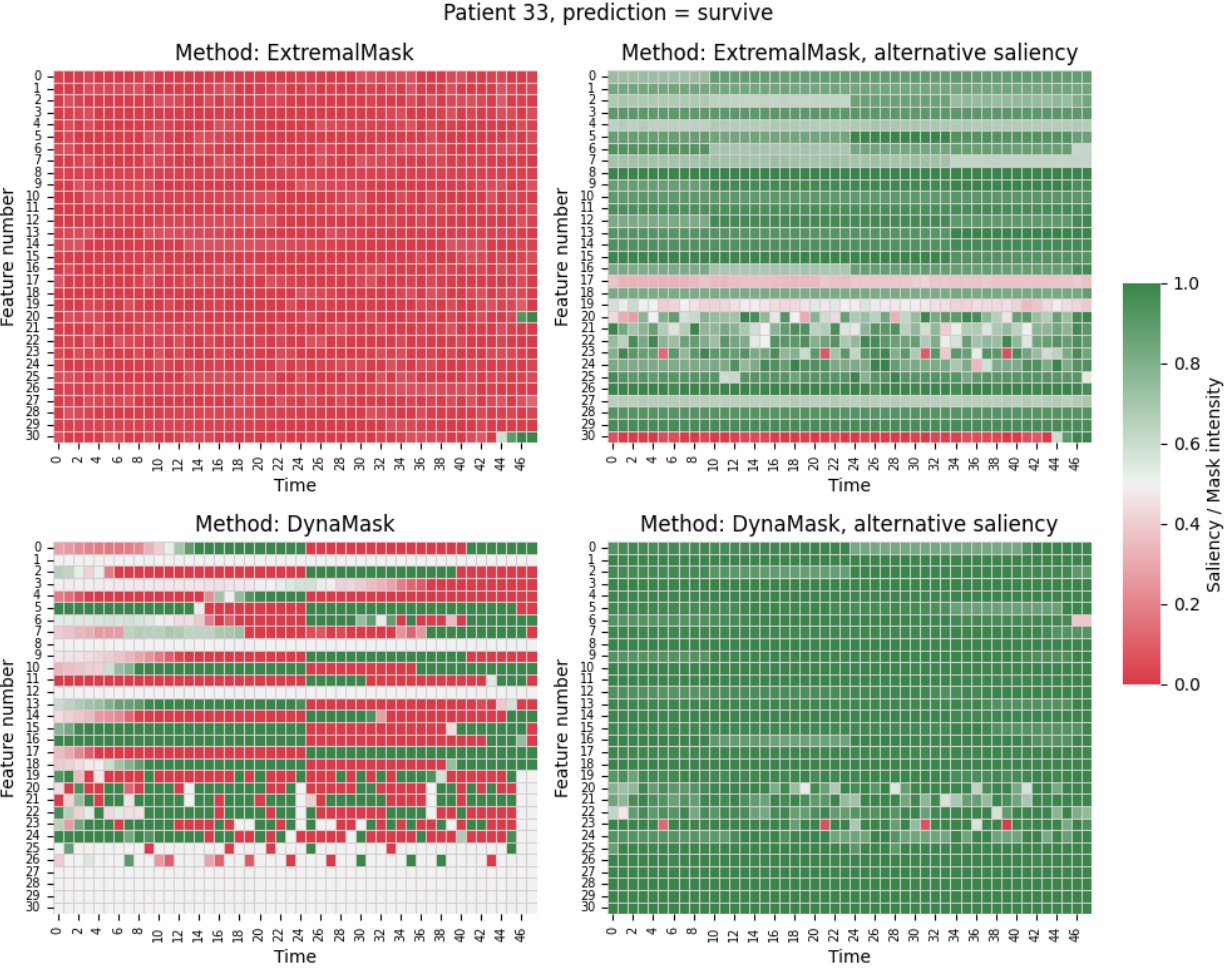

Figure 7: The two left-hand side plots of this figure show original mask saliencies retrieved for ExtremalMask and DynaMask methods. The right-hand side plots of this plot show alternative saliency. These masks are shown for a correctly predicted patient who survived. The more green the point, the more important it was for $f$'s prediction for this sample.

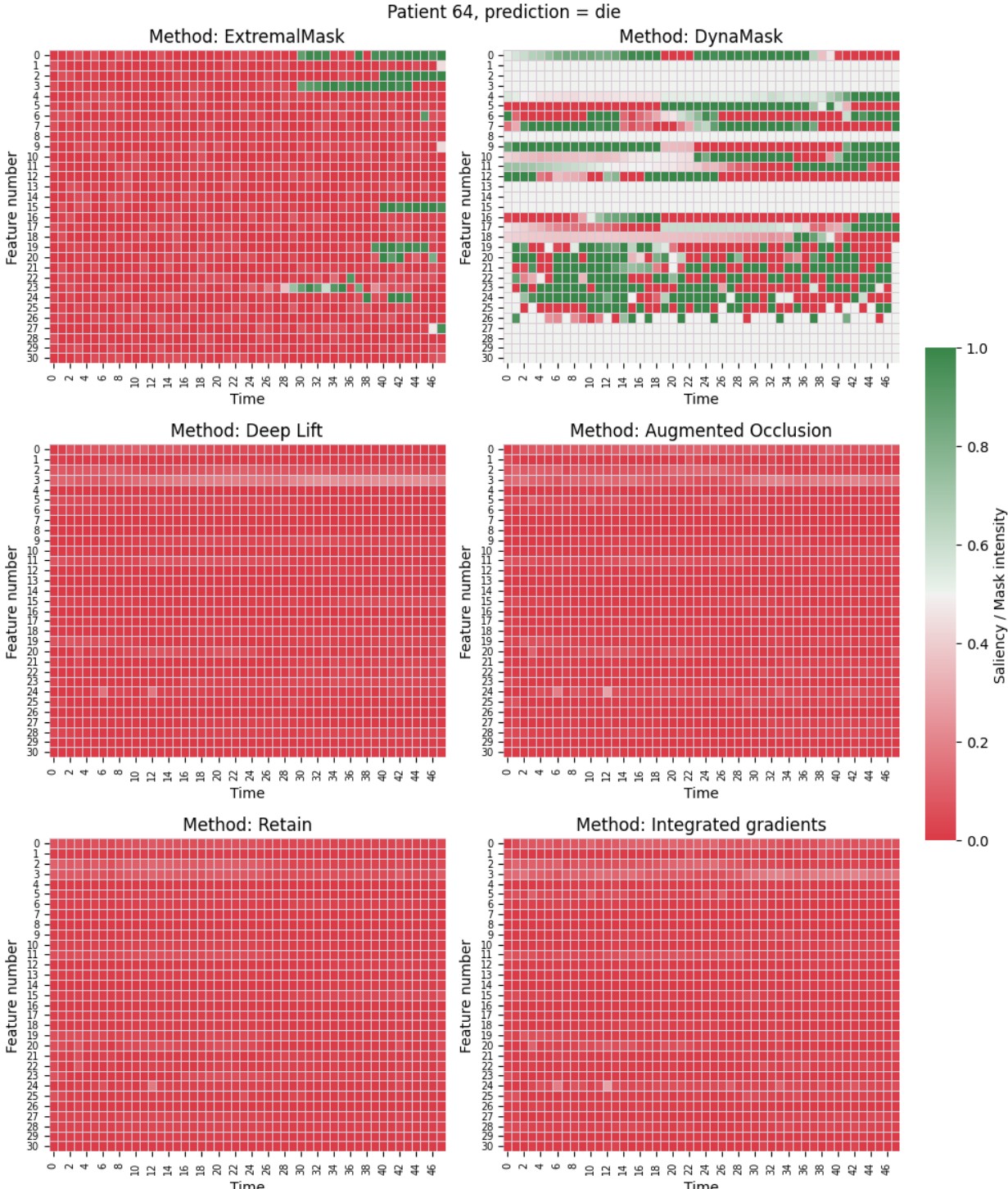

Figure 8: Comparison of the mask-only saliency results from different methods. We show sample 64 from MIMIC-III, seed = 42, fold = 1. The patient was correctly predicted by the classifier $f$ to die. The more green the point, the more important it was for $f$'s prediction for this sample.

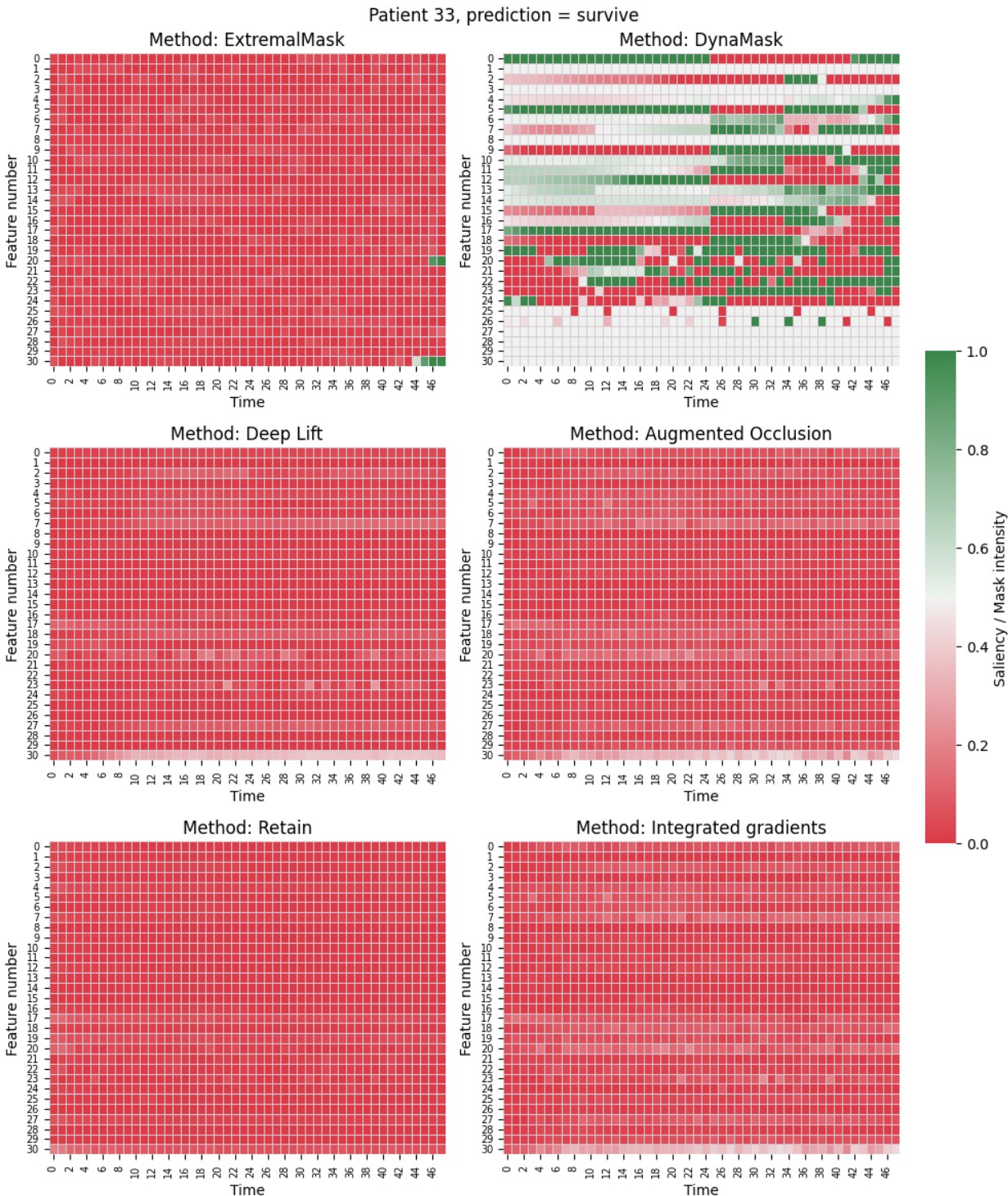

Figure 9: Comparison of the mask-only saliency results from different methods. We show sample 64 from MIMIC-III, seed = 42, fold = 1. The patient was correctly predicted by the classifier $f$ to survive, the more green the point, the more important it was for the classifcation.

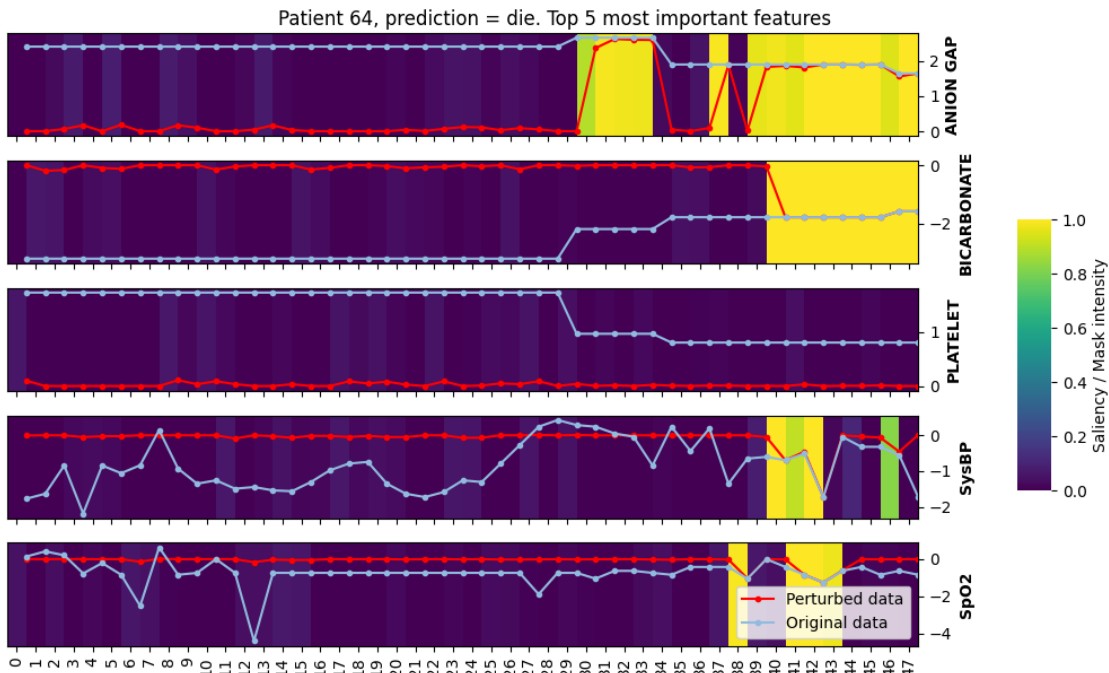

Figure 10: Detailed plot showing the original data, predicted perturbation by ExtremalMask, and the predicted mask-only saliency by ExtremalMask. The results are shown for top 5 most important features as identified by Figure 2

.

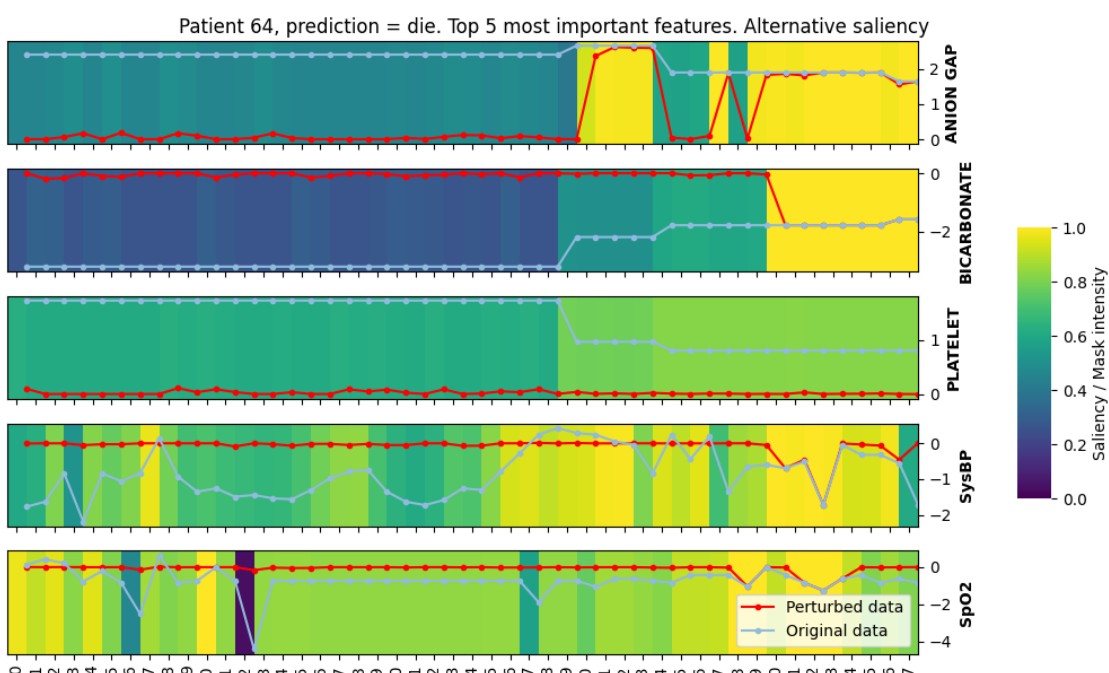

Figure 11: Detailed plot showing the original data, predicted perturbation by ExtremalMask, and the alternative saliency as computed in Equation 4. The results are shown for top 5 most important features as identified by Figure 2

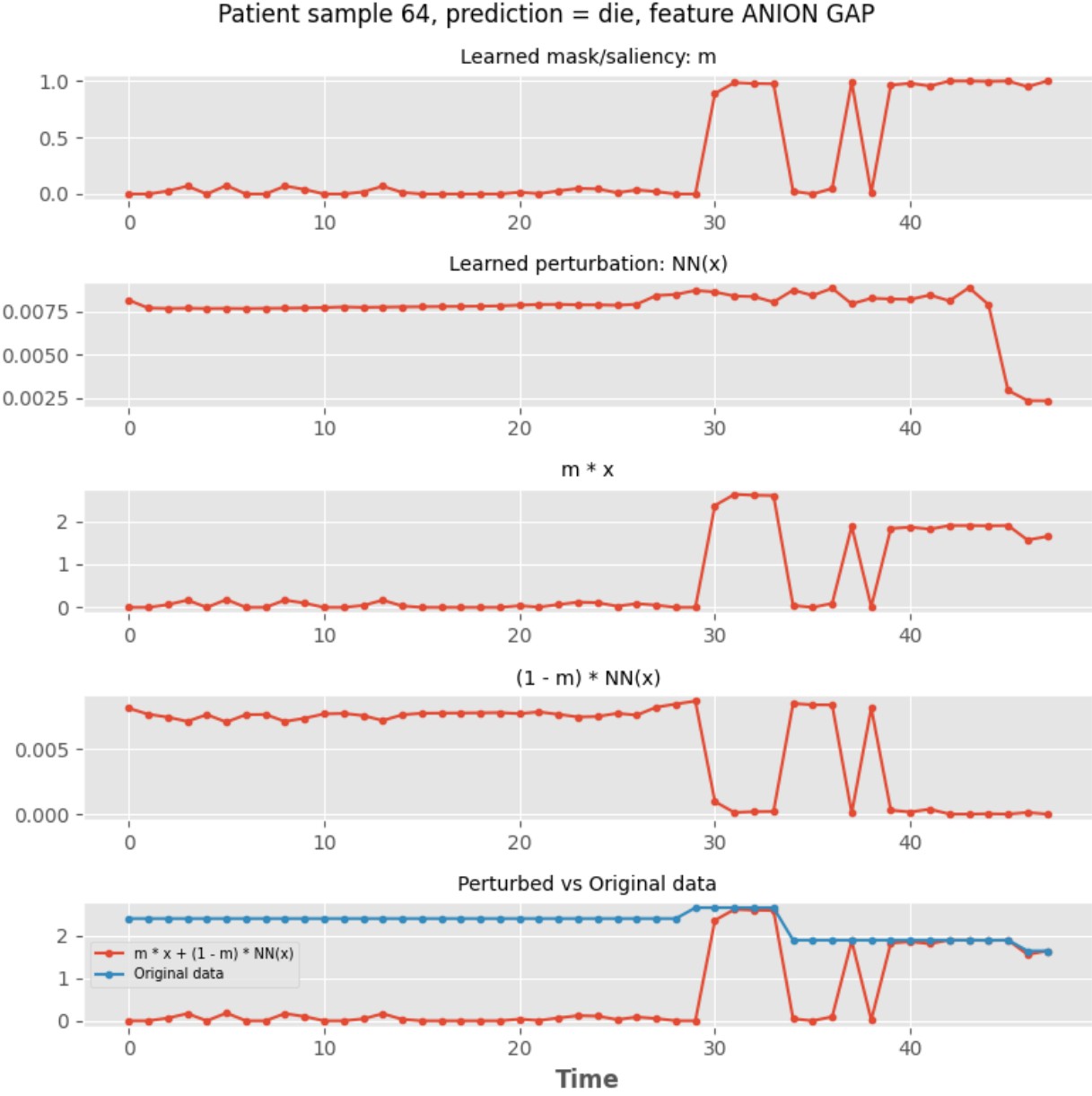

Figure 12: Visualization of Equation Equation 2's components. As in the main text, the 64th sample from MIMIC-III was used with seed = 42 and fold = 0. We focus on the 2nd feature (as e.g. seen in Figure 1a and Figure 10).

### A.7 Visualizations of whitebox HMM experiments

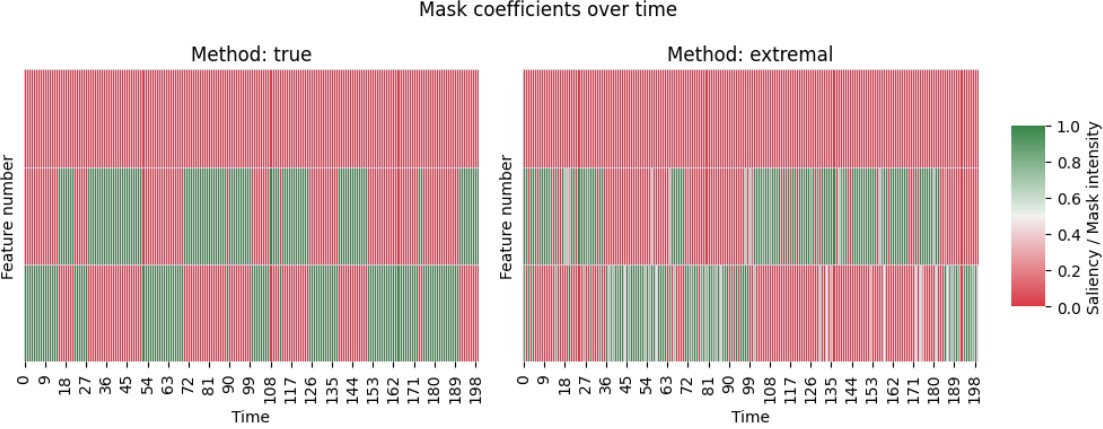

Figure 13: True and predicted saliency by ExtremalMask for HMM dataset. We show sample 12 for seed = 42 and fold = 0.

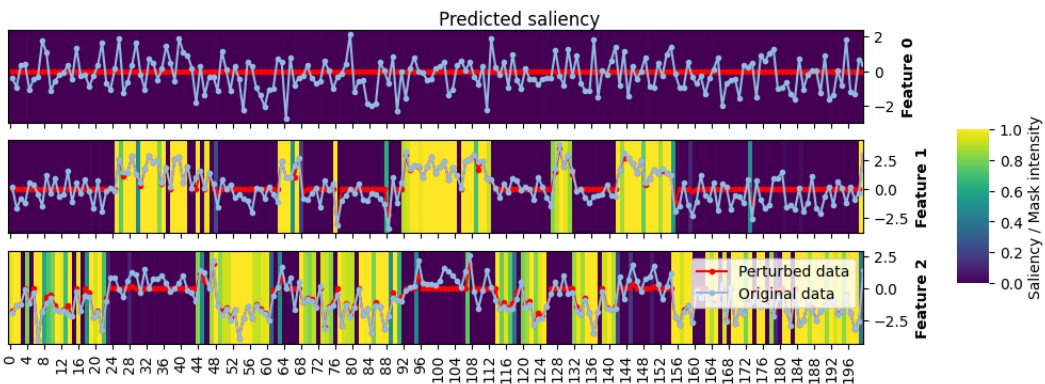

Figure 14: Predicted saliency and perturbation by ExtremalMask for HMM dataset. We show sample 12 for seed = 42 and fold = 0.

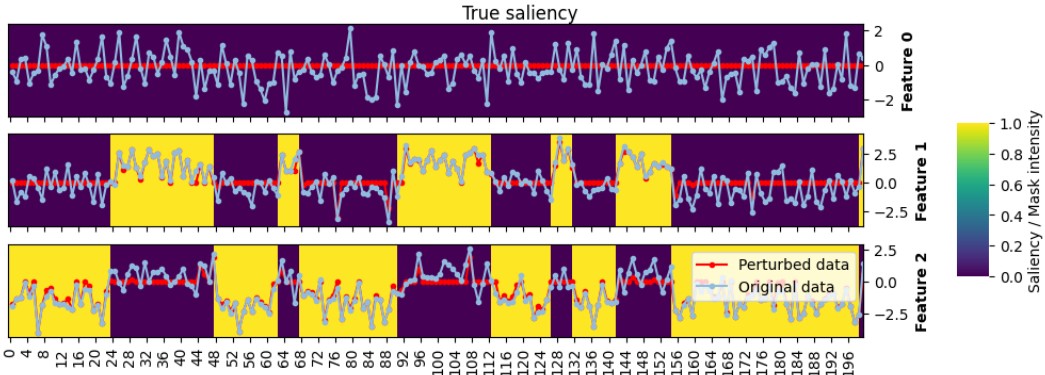

Figure 15: True saliency and perturbation by ExtremalMask for HMM dataset. We show sample 12 for seed = 42 and fold = 0.

