# OpenReview forum: "Reproducibility Study of "Learning Perturbations to Explain Time Series Predictions""
_TMLR — Accepted by TMLR_

### Review · Reviewer_GHn8 · 2024-03-28

**Summary Of Contributions:**

This manuscript is a reproducibility study of Enguehard’s 2023 Learning Perturbations to Explain Time Series Predictions. The authors found general agreement on synthetic and real datasets with the original work and provide novel visualizations of the results and offer nuanced interpretations of the results. They also contribute an alternative metric for assessing saliency and extend the experiments to investigate how practical ExtremalMask is.

**Audience:**

Yes

**Broader Impact Concerns:**

No broader impact concerns

**Claims And Evidence:**

Yes

**Requested Changes:**

Questions:
- In the original paper there were 5 folds run. In this reproducibility study there were only 3. One of the metrics reported was the difference normalized by the standard deviation of the metric. How meaningful is this if the standard deviation is being estimated with only 3 samples? (even if there were 5 would there be much reason to trust the standard deviation estimate very much?)
- “To replicate the results in the original paper, we adopt the default hyperparameters provided by the tint library” Are these the same hyperparameters that were used in the original paper?
- How was 20% chosen as the threshold for saliency when selecting features? Wouldn’t the appropriate thresholding value be dataset dependent?
- In what setting would you expect the newly introduced saliency metric to be more informative than the one used in the original Enguehard paper? In what settings would it be less informative?
- I would appreciate more discussion of the results shown in Figure 2. The values for the alternate metric tend to be higher than for the mask. Should there be equal attribution in totality (across all features) across both metrics? Apologies if I am misunderstanding the plot/attribution

Minor:
- Figure 2 is a little bit difficult to read. The lines overlap each other so the ranges of values is not always clear/easy to read at a glance. The font in the figure is also small
- Occasional typos, grammatical errors

**Strengths And Weaknesses:**

Strengths:
Authors adequately provide context of the main paper before the methods, results, experiments
Provides code for the redone experiments as well as the newly run experiments

Weaknesses:
See requested changes

---

> ### Author Response · Authors · 2024-04-07
>
> Thank you for your feedback! We have edited our paper with the following changes:
>
> 1) As suggested, we reran MIMIC on 5-folds. Furthermore, with a relatively small number of folds, it indeed becomes difficult to draw any kind of conclusion based on the standard deviation. However, it may serve as reference when comparing it with the standard deviations present in the original paper.
>
> 2) 5 days ago, We e-mailed the authors of the original paper about whether the default parameters in the tint library are the ones used in the original paper. Until now, we have not received an reply. We will update the paper once we get a response.
>
> 3) We added that the data with top 20% highest mask values are considered as salient is because this is also done in the original paper. We also added the reason of using 20% in Figure 1b is to be consistent with the original paper. Nevertheless, we decided to repeat the experiments for claim 2 on different saliency thresholds (10%, 30%, 40%,50% and 60%) with the results plotted in Figure 6 in the Appendix. We found that all these results supported claim 2.
>
> 4) This is a good question. We have added the following two sentences to where alternative saliency is introduced in the methodology:
> "both the noise being arbitrarily close to or being distant from $\mathbf{X}_{ntd} \in \mathbb{R}$ could result in low values of the mask in the preservation game."
> "Note that compared to the mask, we would always expect alternative saliency to be at least as informative about the perturbation size, and therefore also at least as informative about the learned saliency of the data by \extrm."
>
> 5)  Please see the following paragraph that is present in the current version of our paper in analyzing figure 2:
> "One would expect that lower mask values lead to larger perturbations and lower alternative saliency values. In turn, this would result in the mask being similar in value as alternative saliency. The observed discrepancy, however, is justifiable when the noise (output of NN) learned by ExtremalMask is, on average, close to the original data, limiting the impact of the mask on the size of the perturbations."
>
> 6) We have made figure 2 more readable, as suggested, and fixed all the typos and grammar errors we found.

---

### Review · Reviewer_6YMW · 2024-03-28

**Summary Of Contributions:**

The authors present a reproducibility study of the paper “Learning Perturbations to Explain Time Series Predictions” with detailed performance evaluations and extensions that develop many areas of the original paper, such as the metric, distribution analysis of the proposed perturbations, and potential future research directions. During the reproduced performance evaluations, the authors were able to confirm with the original authors mistaken reports in the original paper, and provided valuable insights with a comparative study between the original and proposed saliency metric.

**Audience:**

Yes

**Claims And Evidence:**

Yes

**Requested Changes:**

More detailed explanations of the components in the Figure 1 visualizations would strengthen the work.

**Strengths And Weaknesses:**

Strengths:
- The authors reproduce the experiments in the original paper with an attempt to facilitate fair comparison with other methods. For example, in Claim 1, the proposed method (ExtremalMask) and DynaMask were tested after training with CE loss, although only MSE loss was provided in the original code.
- The three claims made from the original paper were well supported through the reproduced experimental results and detailed explanations. The major discrepancies were also well addressed, such as the information and entropy differing by two orders of magnitude being reported to the authors to update the values.
- The two extensions are very important additions to the original paper, because the first one conducts a comparative study between the proposed and original saliency metrics, and the second one conducts experimentation on if the perturbed data of ExtremalMask is realistic. The qualitative analysis in both extensions are easy to follow and supports the effectiveness of each extension.

Weaknesses:
- The visualizations in Figure 1 are not straight-forward to understand, although it can be inferred that the Alternative saliency is more informative than the mask. More detailed explanation on what the visualizations depict would be helpful.
- Although there were resource limitations mentioned by the authors, it is still important to match all experimental settings with the original paper. Therefore, like the original paper which conducted five folds, the authors should have conducted five folds instead of three.

---

> ### Author Response · Authors · 2024-04-07
>
> Thank you for your feedback! As suggested, we reran the experiment on the MIMIC-3 dataset with 5 folds. We have also changed the explanation for figure 1 and the corresponding methodology section to make it more straight-forward to understand.

---

### Review · Reviewer_6VRP · 2024-04-01

**Summary Of Contributions:**

This paper provides a detailed reproducibility study of a particular recent method which optimizes over sequence perturbations to highlight influence in time-series predictions of neural networks. The authors establish an explicit set of claims by the authors of the original work and test each of these claims, as well as experimenting with a few extensions/improvements of their own. They conclude with a summary of the important takeaways as well as a discussion of which elements of the reproduction were easy/difficult and why.

**Audience:**

Yes

**Claims And Evidence:**

Yes

**Requested Changes:**

None, see recommendation above.

**Strengths And Weaknesses:**

Though I've never reviewed a reproducibility study before, this paper comes across to me as a model for how to do it. The original paper's claims are clearly laid out and thoroughly tested. The setup and methodology is carefully described, along with the original motivation. Where the authors' design choices deviate from this, they give clear reasoning for each modification they make to the to original method/evaluation. They further propose several extensions or possible improvements, and they run ablations to compare.

I was especially impressed to see a conclusion which laid out very clearly which elements of the reproduction offered difficulty and why, as this seems invaluable to someone who may be trying to reproduce the idea themselves.

In terms of weakness, I would only give one minor word of caution: some of the language in the expository sections can be a little loose and informal when discussing precise mathematical equations. Unfortunately I think this is not uncommon in the explainability literature, but I recommend the authors take care to try to avoid this. As one example, see the discussion after Equation (2). The first sentence is quite vague, and the "justification" later on feels very hand-wavy.

---

> ### Author Response · Authors · 2024-04-07
>
> Thank you for your feedback! We have added a more rigorous explanation of the assumptions underlying the original paper's implementation in the model description section.

---

### Review · Reviewer_xnfk · 2024-04-02

**Summary Of Contributions:**

This is a reproducibility study of an algorithm that optimizes over sequence perturbations to identify influence in time-series predictions done by neural nets. The authors reproduce the original work, and experiment with a few extensions they propose, and list the key implications of the research and explain what was more difficult to reproduce.

**Audience:**

Yes

**Claims And Evidence:**

Yes

**Requested Changes:**

Seems like very good quality reproduction to me. However, I like the comparison with the choices made in the original paper, but perhaps move that a bit earlier?

**Strengths And Weaknesses:**

This is a high quality reproduction, and the discussion and motivation (and the details) of the original work are well-presented. I like the additional ablations, and discussion of the modifications made. I like the intuition presented behind various formulations.

Overall, the topic is of course interesting, and the paper adds some nice value beyond the original work.

The writing is high quality, though I sometimes feel the emphasis is on intuition rather than mathematical rigor (especially in the early part).

---

> ### Author Response · Authors · 2024-04-07
>
> Thank you for your feedback! We have added some more explanation in the model description to make the math more rigorous.
>
> For the requested changes of moving comparison of choices made in the original paper a bit earlier, we are unsure if it is a good idea as it seems natural to us to introduce the original/general problem first and then transition into a transformed/specific one that is justified under a set of assumptions.

---

### Decision · Action_Editor_gg3r · 2024-05-13

**Recommendation:** Accept as is

**Comment:**

This paper attempts to reproduce the results in an earlier paper. The results obtained by this paper support the claims in the original paper. The paper was reviewed by 4 reviewers, who are all positive after the rebuttal.

**Audience:**

The paper can be of interest to those working on time series prediction.

**Claims And Evidence:**

This paper attempts to reproduce the results in the paper "Learning Perturbations to Explain Time Series Predictions". The authors' results generally agree with those in the original paper. The paper also provides new visualizations and metrics and some new results. The conclusions in the paper are well supported by clear evidences.